**Cite this article:** de Jonge DSW, Merten V, Bayer T, Puebla O, Reusch TBH, Hoving H-JT. 2021 A novel metabarcoding primer pair for environmental DNA analysis of Cephalopoda (Mollusca) targeting the nuclear 18S rRNA region. *R. Soc. Open Sci.* **8**: 201388. https://doi.org/10.1098/rsos.201388

molecular biology/oceanography

metabarcoding, environmental DNA, Cephalopoda, universal primer

**Author for correspondence:**
Daniëlle S. W. de Jonge
e-mail: dsd3@hw.ac.uk

†Present address: The Lyell Centre for Earth and Marine Science and Technology, Heriot−Watt University, Edinburgh EH12 4AS, UK.

# A novel metabarcoding primer pair for environmental DNA analysis of Cephalopoda (Mollusca) targeting the nuclear 18S rRNA region

Daniëlle S. W. de Jonge[1,†], Véronique Merten[2], Till Bayer[2], Oscar Puebla[2,3], Thorsten B. H. Reusch[2] and Henk-Jan T. Hoving[2]

[1]Faculty of Mathematics and Natural Sciences, University of Groningen, Groningen, The Netherlands
[2]Marine Evolutionary Ecology, GEOMAR Helmholtz Centre for Ocean Research Kiel, Kiel, Germany
[3]Ecology Department, Leibniz Centre for Tropical Marine Research (ZMT), Bremen, Germany

DSWdJ, 0000-0002-4093-2721; VM, 0000-0003-1154-1585; TB, 0000-0002-4704-2449; OP, 0000-0001-9700-5841; TBHR, 0000-0002-8961-4337; H-JTH, 0000-0002-4330-6507

Cephalopods are pivotal components of marine food webs, but biodiversity studies are hampered by challenges to sample these agile marine molluscs. Metabarcoding of environmental DNA (eDNA) is a potentially powerful technique to study oceanic cephalopod biodiversity and distribution but has not been applied thus far. We present a novel universal primer pair for metabarcoding cephalopods from eDNA, *Ceph18S* (Forward: 5′-CGC GGC GCT ACA TAT TAG AC-3′, Reverse: 5′-GCA CTT AAC CGA CCG TCG AC-3′). The primer pair targets the hypervariable region V2 of the nuclear 18S rRNA gene and amplifies a relatively short target sequence of approximately 200 bp in order to allow the amplification of degraded DNA. *In silico* tests on a reference database and empirical tests on DNA extracts from cephalopod tissue estimate that 44–66% of cephalopod species, corresponding to about 310–460 species, can be amplified and identified with this primer pair. A multi-marker approach with the novel *Ceph18S* and two previously published cephalopod mitochondrial 16S rRNA primer sets targeting the same region (Jarman *et al.* 2006 *Mol. Ecol. Notes.* **6**, 268–271; Peters *et al.* 2015 *Mar. Ecol.* **36**, 1428–1439) is estimated

to amplify and identify 89% of all cephalopod species, of which an estimated 19% can only be identified by *Ceph18S*. All sequences obtained with *Ceph18S* were submitted to GenBank, resulting in new 18S rRNA sequences for 13 cephalopod taxa.

# 1. Introduction

Cephalopods, the molluscan class to which squids, octopods, cuttlefish and vampire squids belong, occur in all the world's oceans from the intertidal zone to the deep sea [1–3]. Their high protein content and large populations make them important in commercial fisheries and food–web interactions as both predator and prey [4–8]. Cephalopods are among the giants of the ocean (giant squid *Architeuthis* spp., colossal squid *Mesonychoteuthis hamiltoni*), and the highest diversity at the family level is found in the deep sea [9].

Deep-sea cephalopods have evolved specialized traits to cope with life in continuous darkness, but basic biological data are still lacking for the majority of species. One of the reasons for this paucity of information is that cephalopods are difficult to study with traditional sampling methods like net catches or video surveys [2]. The size of trawled cephalopods is biased by the used mesh and net size and specimens often get entangled and damaged in the mesh which may make it difficult to identify them morphologically [10]. Additional difficulties in sampling cephalopods result from their possibly patchy distribution and their agility which allows them to avoid or escape sampling gear. Video surveys with submersibles and towed cameras require lights that are easily detectable by the well-developed cephalopod eyes, which may result in avoidance behaviour [3]. The study of cephalopod remains in the stomachs of predators provides indirect evidence of their presence [11–13], but the digested state of cephalopod remains may hamper species identification, and the selectivity of the predators and limited knowledge of their foraging area introduces bias. The challenges associated with cephalopod sampling, particularly in remote areas such as deep pelagic environments, raise an urgent need for novel monitoring methods.

Environmental DNA (eDNA) analysis constitutes a promising tool to study the distribution and diversity of cephalopods. This technique is based on the idea that organisms leave DNA in the environment, and that this DNA can be extracted and sequenced to identify the species from which it originates [14]. PCR amplification of eDNA in a sample (e.g. filtered from water) can detect the presence of a species or of multiple taxa by targeting a variable gene with either a species-specific or universal primer set, respectively. Biodiversity assessments of an eDNA sample using a universal primer set fall under the broader term metabarcoding [15,16], i.e. parallel identification of multiple taxa from one complex DNA sample. Metabarcoding has its origins in microbiology, palaeoecology and diet analysis [17–21]. Sampling eDNA is relatively easy in the marine environment (from water or sediment) and developments in the field of next-generation sequencing have greatly reduced sequencing costs and dramatically increased sequencing output. Therefore, eDNA analysis represents a non-invasive and cost-effective method for biodiversity assessments, especially for rare or elusive species and in remote areas [14,22–25].

Metabarcoding of eDNA from seawater has mostly been used to identify fishes [26–28] and assess overall (metazoan) eukaryotic diversity [29–32], but to our knowledge has not been used to focus on specific taxonomic groups like cephalopods. Metabarcoding has only been applied to cephalopod eDNA in studies focused on larger taxonomic groups from coastal areas, e.g. eukaryotes [29], metazoans [31] and invertebrates [33]. Even though these studies did not specifically focus on cephalopod distribution, they were able to provide valuable insights on cephalopod distribution. For example, the distribution of *Sepiola tridens*' eDNA suggested that the species' distribution might extend further into the coastal zones of Northern Europe than previously thought [31]. In recent years, there has been a sharp increase in the number of studies that employed eDNA analysis in the deep sea [22,34,35], but none to specifically investigate cephalopod diversity. Recently, a species-specific primer has been developed and used to detect the giant squid, *Architeuthis dux*, in the photic zone of the Sea of Japan [25].

Universal metabarcoding primers, i.e. a single primer set that targets multiple taxa, should have a common annealing site in all taxa and amplify a sequence with enough variation to distinguish between groups at the desired taxonomic resolution, which often is the species level [36]. Ideally, a pair of universal primers will target the largest possible taxonomic group of interest, unambiguously identify all species, while not amplifying non-target taxa. However, due to the often degraded state of eDNA, a target sequence size of less than 300 bp, sometimes referred to as a mini-barcode [37],

is preferable [36]. Therefore, the primer pair that amplifies approximately 650 bp of the cytochrome-oxidase-1 (COI) [38,39], which is most commonly used for species identification, is unsuitable for eDNA metabarcoding. In practice, there is a trade-off between the taxonomic range of amplifiable species (i.e. universality), indicated by the coverage index ($B_c = n.$ of amplified taxa/$n.$ of target taxa), and the resolution for identification at species level, indicated by the specificity index ($B_s = n.$ of identified taxa/$n.$ of amplified taxa) [40]. The analysis of eDNA using multiple markers, preferably from different genes, can increase the sensitivity of the technique and overcome the specificity issues of a single pair of universal primers [29]. So far, two sets of universal primers specifically targeting cephalopods have been published, one targeting the mitochondrial 16S rDNA [41,42] and one targeting the cytochrome $b$ region [43]. However, the latter is not suitable for eDNA metabarcoding due to the long target amplicon size. This study describes the development of a novel set of universal primers for cephalopods targeting the nuclear 18S rRNA region and describes its complementarity to the mitochondrial 16S rDNA primer set [41,42] in a multi-marker approach to study the diversity and distribution of cephalopods through metabarcoding of eDNA.

# 2. Material and methods

The process for the development of a metabarcoding primer pair encompassed four steps: (i) assembling a reference database, (ii) identifying potential primer sets, (iii) *in silico* testing, and (iv) empirical testing.

## 2.1. Reference database

Two cephalopod 18S rRNA databases were generated from GenBank and SILVA. The results for a general GenBank query '18S Cephalopoda' included environmental samples and partial sequences from other 18S subregions. To ensure exclusion of environmental samples and inclusion of sequences from the same subregion, the first sequence of the largest subset of partial sequences (the longfin inshore squid *Doryteuthis pealeii*, MH586846, 760 bp) was aligned against the full GenBank database (discontinuous megablast, 9 October 2018) [44] and all matches with full query cover were downloaded. This approach resulted in 31 partial cephalopod 18S rRNA sequences from 24 species. For the SILVA database, all cephalopod 18S rRNA reference sequences were downloaded (11 October 2018) and included 146 sequences from 88 species ranging from 423 to 2610 bp. The NCBI taxonomy dump file was downloaded (18 September 2018) and used for sequence annotation of both the GenBank and SILVA database with unique taxonomic identifications using Python v. 2.7.15 and OBITools v. 1.2.10 [45]. The annotated extended Fasta files were converted to the reference ecoPCR database v. 0.2 format [40].

## 2.2. Identification of potential primer sets

The ecoPrimer v. 0.3 algorithm [16] was used to identify potential primer pairs of 20 bp each that amplify a target sequence of 50–200 bp [46] with ample variation for taxonomic resolution to species level. The settings used on the GenBank reference database did not return any potential primer sets when applied to the SILVA reference database; hence for the SILVA database, the settings were somewhat relaxed (algorithm parameters in table 1). The potential primers were filtered for lowest melting temperature ($T_m$) between 59 and 69°C, maximum difference between lowest melting temperatures less than 3°C, GC count of 50–60%, a GC clamp with less than four G and/or C, less than four nucleotide repeats and less than four dinucleotide repeats [46]. The relaxed settings for the SILVA database may have resulted in suboptimal primers. Therefore, only primer sets with a $B_c$ equal to or higher than the GenBank derived primer sets when tested *in silico* against the SILVA database were considered and filtered for lowest melting temperature ($T_m$) (between 45 and 70°C) and a GC count between 45 and 65%.

## 2.3. *In silico* testing

From the filtered list, eight primer sets with the highest coverage index ($B_c$) and specificity index ($B_s$), as estimated by ecoPrimer [16], were chosen for further tests *in silico*, i.e. to predict their effectiveness in PCR amplification. The primers were analysed for secondary structures (hairpins and primer-dimers) using the online Oligonucleotide Properties Calculator v. 3.27 [47], and for self-complementarity and *in silico* amplification using online Primer-BLAST [48]. The $B_c$ and $B_s$ of the primer set was calculated with an ecoPCR v. 0.2 [40] *in silico* test against the SILVA database (no mismatches allowed). Three primer sets

**Table 1.** Parameters settings used for the ecoPrimer v. 0.3 analysis [16]. GB, GenBank reference database (the analysis was run twice with different mismatch and 3′ match settings). SV, SILVA reference database (quorum parameters were relaxed relative to the GenBank settings).

| parameter | value | description |
| --- | --- | --- |
| primer length (bp) | 20 | required length of forward and reverse primer |
| target amplicon length (bp) | 50–200 | required length range of the amplified sequence |
| strict matching quorum | GB: 0.7 | minimum fraction of sequence records in the reference database |
| | SV: 0.5 | with an exact match between the primer and target sequence |
| sensitivity quorum | GB: 0.9 | minimum fraction of sequence records in the reference database |
| | SV: 0.7 | that exactly match the specified parameters |
| no. mismatches | GB: 0 and 3 | number of allowed mismatches between primer and target sequence |
| | SV: 3 | |
| no. 3′ matches | GB: NA and 2 | number of strict matches required at the 3′ end |
| | SV: 2 | |

with optimal characteristics, i.e. no or limited secondary structures and highest $B_c$ and $B_s$ indices, were ordered for empirical testing.

The $B_c$ and $B_s$ of the two mitochondrial 16S rRNA primer sets, CephMLS (CephMLSf1: 5′-TGC GGT ATT WTA ACT GTA CT-3′, CephMLSr1: 5′-TTA TTC CTT RAT CAC CC-3′) [41] and S_Cephalopoda (S_Cephelapoda-F: 5′- GCT RGA ATG AAT GGT TTG AC-3′; S_Cephalopoda-R: 5′-TCA WTA GGG TCT TCT CGT CC-3′) [42] were estimated to determine the complementarity of the newly developed and existing primer sets. The target sequence size of S_Cephalopoda is 70–73 bp [42] and falls right within the targeted region of CephMLS, which has a target sequence size of 212–244 bp [41]. A 16S reference database was obtained by using Primer-BLAST [48] with the respective primer sets, and subsequently blasting the first match (blastn protocol, limit to cephalopod taxon in nucleotide database, exclude uncultured/environmental samples, query cover greater than 50%) [44] to obtain sequences that were known to contain the targeted region. An ecoPCR v. 0.2 [40] in silico test (no mismatches allowed) was used to determine $B_c$ and $B_s$ indices for S_Cephalopoda and CephMLS using this 16S GenBank reference database.

The 18S primer development process was based on two reference databases: one from SILVA with 146 sequences, and one from GenBank with 31 sequences. The latter has significantly less sequences than the SILVA database, caused by our specific filtering choices to avoid non-overlapping sequences which would have obstructed the development process. We calculated the $B_c$ and $B_s$ indices for our new primer set from the SILVA database during the development process. However, we felt that a comparison between these SILVA-derived 18S indices and the Primer-BLAST-derived 16S indices would be biased. SILVA is specific about which GenBank sequences are admitted into the alignment, and some GenBank sequences might have been left out. To ensure an unbiased comparison between 16S and 18S coverage and specificity indices, we obtained a third 18S database by using the newly developed 18S primer sequence in a GenBank Primer-BLAST [48]. This third GenBank database could not have been obtained at the start when we had not yet developed the primer set.

## 2.4. Empirical testing

Cephalopod specimens were collected in the eastern tropical Atlantic in waters near the Republic of Cabo Verde by the R.v. *Walter Herwig III* in March and April 2015 (cruise ID WH383, permissions obtained from *Ministério da Agricultura e Ambiente* and *Agência Marítima e Portuária* of Cape Verde). The sampling net was a pelagic trawl (Engel Netze, Bremerhaven, Germany, length 18 m, 16 × 30 m mouth opening, cod end 20 mm stretched mesh-opening, 1.8 mm inlet sewn into last 1 m of cod end) with a multi-sampler allowing depth-stratified sampling (electronic supplementary material, table S1). The specimens were morphologically identified by H.-J.T.H., and the full specimen (for small individuals) or a part of an arm (for larger individuals) was stored in a 2 ml tube with ethanol. DNA was extracted from the identified cephalopod tissue samples with the DNeasy Blood and Tissue Kit (Qiagen) following the manufacturer's protocol.

DNA purity and concentration were measured with NanoDrop (Thermo Fisher Scientific) and Qubit (dsDNA broad range assay kit, Thermo Fisher Scientific), respectively. DNA extracts diluted to approximately 10 ng µl⁻¹ of three species from different families (*Bathyteuthis abyssicola*, *Heteroteuthis dispar* and *Liocranchia reinhardtii*), a mixture of these three extracts, negative extraction controls (no tissue) and negative PCR controls, i.e. with PCR grade water instead of DNA template, were amplified with a temperature gradient for all selected potential new primer pairs. A PCR mixture of 40 µl reaction volume of which 10 µl DNA template was prepared with the KAPA Hifi kit (Kapa Biosystems, Roche Inc.) (1× Fidelity buffer [which corresponds to 0.4 mM MgCl$_2$], 0.3 mM KAPA dNTPs, 5% DMSO, 0.02 U µl⁻¹ KAPA Hotstart Polymerase, 0.5 µM of each forward and reverse primer). An Applied Biosystems Veriti Thermal Cycler (Thermo Fisher Scientific) was used for the PCR reaction. The PCR programme consisted of a 5 min initial denaturation step at 95°C followed by 35 cycles of denaturing at 98°C for 20 s, annealing temperature gradient for 15 s and extension at 72°C for 1 min, followed by a final elongation step at 72°C for 10 min and 4°C on hold. The temperature gradient started at 3°C below the lowest $T_m$ of the primer set and increased with five steps of 3°C each. This temperature gradient was chosen to account for the expected increase in optimal annealing temperature due to the KAPA Hifi kit, and an expected decrease in optimal annealing temperature due to the DMSO in the PCR mix, which together could cause deviation from the theoretical optimal annealing temperature by several degrees Celsius. The PCR products were visualized under UV light on a 1.2% agarose gel using loading dye, GelRed (Biotium) and a 100 bp ladder (Thermo Fisher Scientific). Once the optimal annealing temperature was determined, the same PCR procedure was conducted on more cephalopod tissue DNA extracts (30 species, figure 4; electronic supplementary material, table S1), DNA extract mixture of the newly tested species, and negative extraction and PCR controls.

PCR products from the primer set producing the most promising results, i.e. clear bands and limited smearing and non-specific bands on the agarose gel, were used for Sanger sequencing. Sanger sequencing was performed in GEOMAR and at the Institute of Clinical Molecular Biology (IKMB) in Kiel using different protocols. In GEOMAR, the PCR products were prepared for sequencing using the Sanger Sequencing Kit (Applied Biosystems). A reaction volume of 12 µl (10 µl PCR product, 0.03 U µl⁻¹ FastAP, 0.33 U µl⁻¹ ExoI) was used to remove unincorporated primers with the following PCR conditions: 37°C for 20 min, 80°C for 15 min and 4°C on hold. The sequencing reaction was conducted in a volume of 10 µl (0.5 µl cleaned PCR product, 0.5× Sequencing buffer, 2.5% BigDye Terminator Mix 3.1, 0.25 µM forward *or* reverse primer) with the following PCR conditions: 1 min at 96°C, 28 cycles at 96°C for 10 s, primer annealing temperature for 5 s, 60°C for 4 min and hold at 8°C. Finally, the sequencing reaction was purified with a bead-based reagent (62% Sam solution and 13% BigDye XTerminator) by shaking the mixture for 30 min at full speed and centrifuging for 2 min at 1000 r.p.m. At the IKMB, the PCR products were purified in a reaction volume of 10 µl (8 µl PCR product, 0.06 U µl⁻¹ FastAP, 0.30 U µl⁻¹ ExoI) with the following PCR conditions: 37°C for 10 min, 75°C for 15 min and 10°C on hold. The sequencing reaction was conducted in a volume of 10 µl (2 µl purified PCR product, 0.75× Sequencing buffer, 7% BigDye Terminator Mix 3.1, 0.32 µM forward *or* reverse primer) with the following PCR conditions: 1 min at 96°C, 25 cycles at 96°C for 10 s, 50°C for 5 s, 60°C for 4 min and hold at 10°C. The sequencing products were purified through Sephadex G-50 fine gel filtration (GE Healthcare Buchler).

Low-quality ends and primers were trimmed manually from the Sanger sequences, which were then manually checked and edited using 4Peaks v. 1.8 [49], and subsequently assembled using AliView v. 1.24 [50]. The assembled sequences (or single forward or reverse sequences in cases of failed sequencing) were checked against the online GenBank reference database with BLAST (megablast algorithm, nucleotide collection nr/nt, 29 June 2020) [44].

# 3. Results

## 3.1. Potential primer pairs

All three empirically tested primer pairs amplified cephalopod DNA of the expected length. However, two primer pairs produced electrophoresis gels with side products, i.e. smearing and non-specific bands, potentially due to false priming or greater propensity to form secondary structures. The primer pair with the cleanest amplification was named 'Ceph18S' (table 2) and selected for sequencing. The other two alternatives were not tested beyond this point (alternative 1: F-5'-GCA CTT AAC CGA CCG

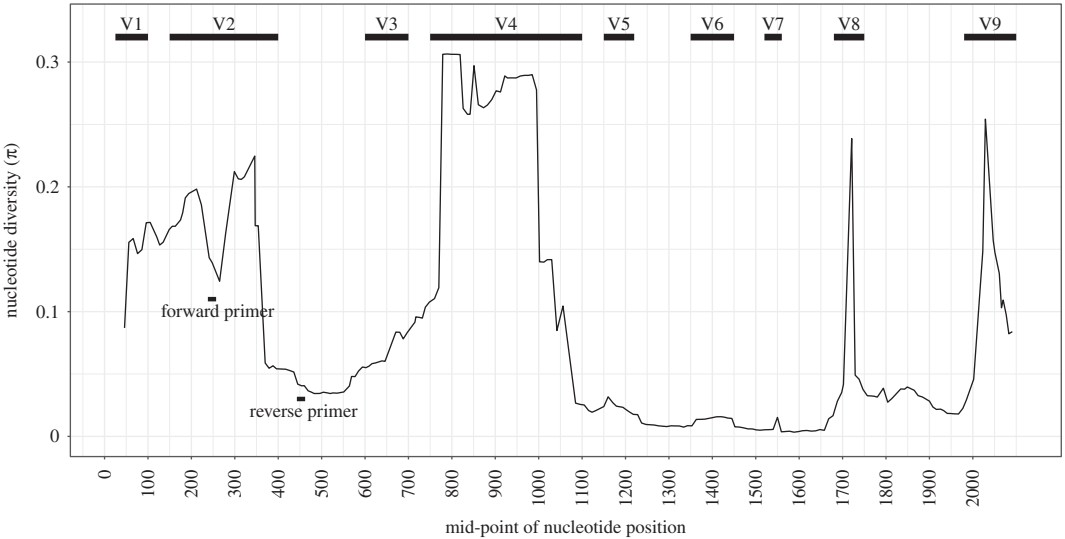

**Figure 1.** Position of the forward and reverse *Ceph18S* primers on the cephalopod 18S rRNA gene. Nucleotide diversity ($\pi$) was calculated with a sliding-window analysis (window size = 99 bp, step size = 10 bp) over the SILVA alignment (after removal of large indels 0–572, 2004–2411, 6255–6529) and shown based on the reference sequence of *L. formosana* (accession code: AY557478).

**Table 2.** Properties of the developed universal cephalopod primer pair targeting the 18S rRNA region to be used in eDNA metabarcoding.

| name | *Ceph18S* |
|---|---|
| forward | 5′-CGCGGCGCTACATATTAGAC-3′ |
| reverse | 5′-GCACTTAACCGACCGTCGAC-3′ |
| forward $T_m$[a] | 59.3°C |
| reverse $T_m$[a] | 61.7°C |
| target length | ~150–190 bp |
| $B_c$[b] | 0.85 |
| $B_s$[b] | 0.78 |

[a]The $T_m$ is estimated using the SantaLucia method [51] for a salt concentration of 0.05.
[b]$B_c$ is *in silico* coverage index (amplified taxa/target taxa) and $B_s$ is *in silico* specificity index (identified taxa/amplified taxa) based on the SILVA database with 146 sequences of 97 unique taxa.

TCG AC-3′ and R-5′-GTC GCG GCG CTA CAT ATT AG-3′; alternative 2: ATT AGA CTG AGA CCG ATG CG-3′ and R-5′-GAC CGT CGA CAG TTG ATA GG-3′).

The *Ceph18S* primer pair targets the V2 variable region of the small-subunit 18S rRNA gene [52] (figure 1). Alignment against *Loligo formosana* (accession code AY557478) shows that the target sequence is located at positions 258–442 (figure 1), but the exact location will depend on the species due to the variation in this region. The *in silico* PCR with *Ceph18S* on the SILVA database shows a target sequence length ranging from 131 bp for *Watasenia scintillans* to 196 bp for *Sepia elegans* (table 2). The alignments of these ecoPCR target sequences and sequences obtained in the laboratory clearly show conserved flanking regions and a highly variable internal region (figure 2). The optimal annealing temperature for *Ceph18S* in the PCR master mix used in this study was found to be 62°C. This differs from the calculated $T_m$ (table 2) as both the KAPA reagents and DMSO in our PCR master mix alter the annealing temperature.

## 3.2. *Ceph18S* resolution

The power of the *Ceph18S* primer pair to identify cephalopod species was examined with an *in silico* PCR on 88 species plus nine unique genera (i.e. genera without species identification which complemented

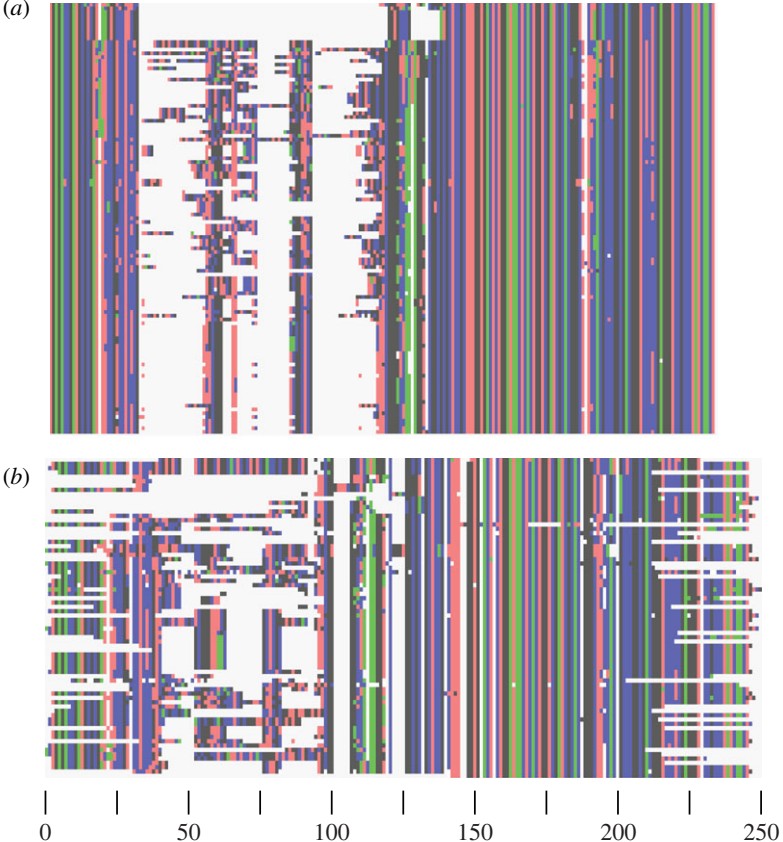

**Figure 2.** Target sequences alignment. Alignment of (*a*) target sequences obtained through *in silico* PCR and (*b*) sequences of extracted cephalopod DNA, both amplified with *Ceph18S*. Alignment using MUSCLE and image creation done in AliView [50].

the genera of the 88 identified species) in the SILVA database. From the 97 unique taxa in the SILVA database, 82 could be amplified *in silico* ($B_c = 0.85$). Of these 82 amplified taxa, 64 could be identified unambiguously ($B_s = 0.78$, figure 3). Therefore, 18 taxa could not be identified unambiguously, i.e. their target sequence was equal to the target sequence of another species: (i) *Leachia atlantica* and *Leachia lemur*, (ii) *Octopoteuthis megaptera* and *Taningia danae*, (iii) *Chiroteuthis veranyi* and *Chiroteuthis calyx*, (iv) *Discoteuthis laciniosa* and *Discoteuthis discus*, (v) *Onykia carriboea* and *Ancistroteuthis lichtensteinii*, (vi) *Selenoteuthis scintillans* and *Lycoteuthis lorigera*, (vii) *Histioteuthis miranda* and *Histioteuthis bonellii*, (viii) *Gonatus antarcticus* and *Gonatopsis* sp., and (ix) *Gonatopsis octopedatus*, and *Okutania anonycha*. Taxa for which some reference sequences were amplified but not all, were *Chtenopteryx sicula, Loligo forbesi, Sepia elegans, Sepiella inermis* and *Todaropsis eblanae.* Further inspection revealed that some reference sequences of these species were incomplete, i.e. omitting at least the V2 region around which the *Ceph18S* primer set anneals. Taxa that were not amplified at all due to a lacking reference V2 region were *Eledone cirrhosa, Euprymna scolopes, Hapalochlaena maculosa, Loligo vulgaris, Octopus vulgaris, Opisthoteuthis* sp. and *Rossia macrosoma.* Taxa that were not amplified even though a reference V2 region was available were *Alloteuthis* sp., *Bathypolypus* sp., *Cirrothauma murrayi, Pyroteuthis margaritifera, Sepia pharaonis, Sepioloidea lineolata, Spirula spirula* and *Vampyroteuthis infernalis.*

Subsequently, *Ceph18S* was tested empirically on 75 tissue DNA extracts (figure 4) from 68 specimens (electronic supplementary material, table S1) representing 30 cephalopod species plus two unique genera from specimens that could not be identified to species level (sequences are deposited in GenBank, accession numbers MT680727–MT680790 and MT680792–MT680800). Of the 32 taxa, 14 taxa did not have corresponding 18S rRNA reference sequences in GenBank (figure 4). All tested taxa, except for *Discoteuthis discus* and *Vitreledonella richardi*, could be amplified with *Ceph18S*, although sometimes with suboptimal sequence quality, so empirical $B_c = 0.94$. The *Ceph18S* barcodes submitted to GenBank add previously non-existent 18S rRNA reference sequences for 13 taxa: *Abraliopsis atlantica, Abralia redfieldi, Bathothauma lyromna, Chiroteuthis* cf. *joubini, Egea inermis, Helicocranchia pfefferi, Heteroteuthis*

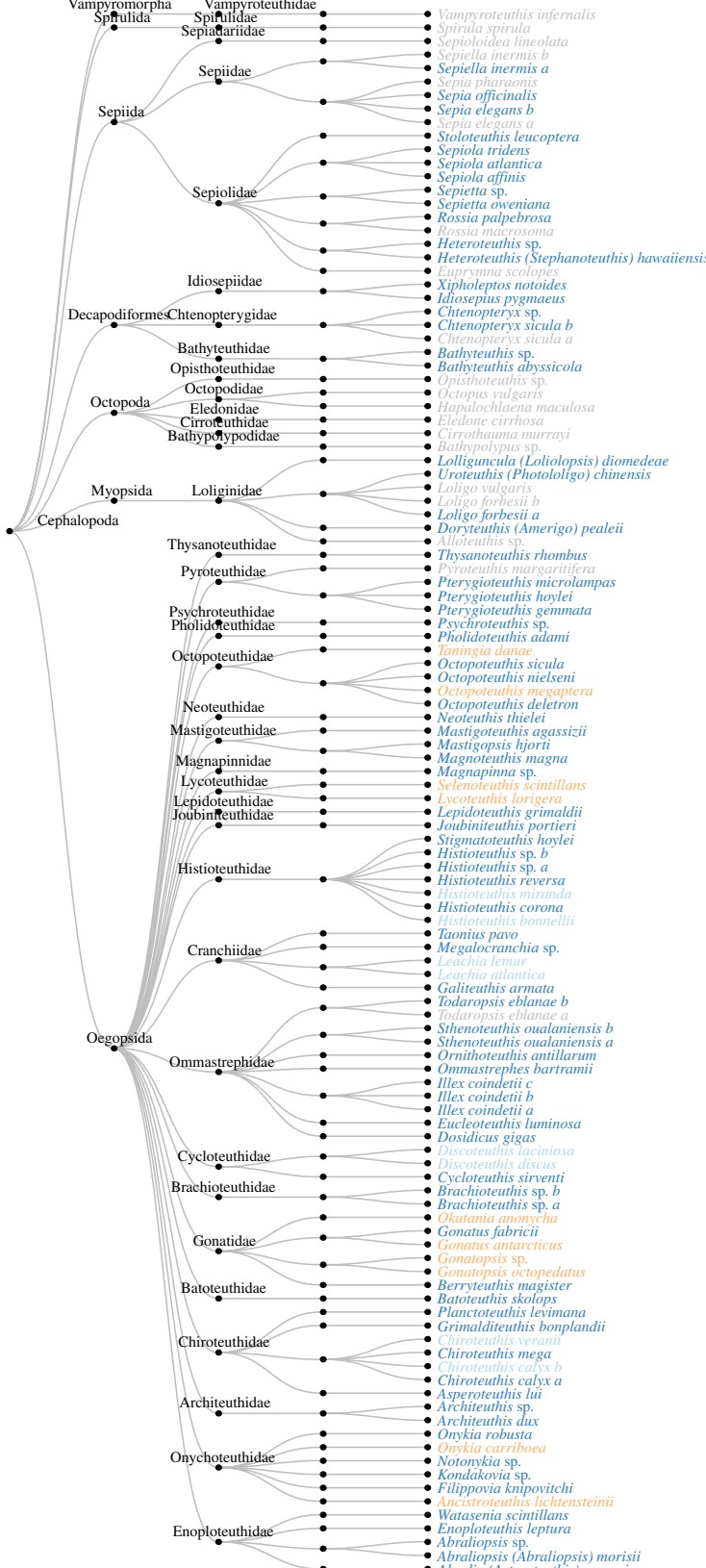

**Figure 3.** Results of *in silico* amplification of the SILVA database (146 sequences, 97 taxa) with *Ceph18S*. Grey taxa were not amplified *in silico*, whereas dark blue taxa were amplified and unambiguously identified. When taxa had equal target sequences and hence could not be unambiguously identified, colour indicates whether the shared taxonomic level was genus (light blue) or family (orange).

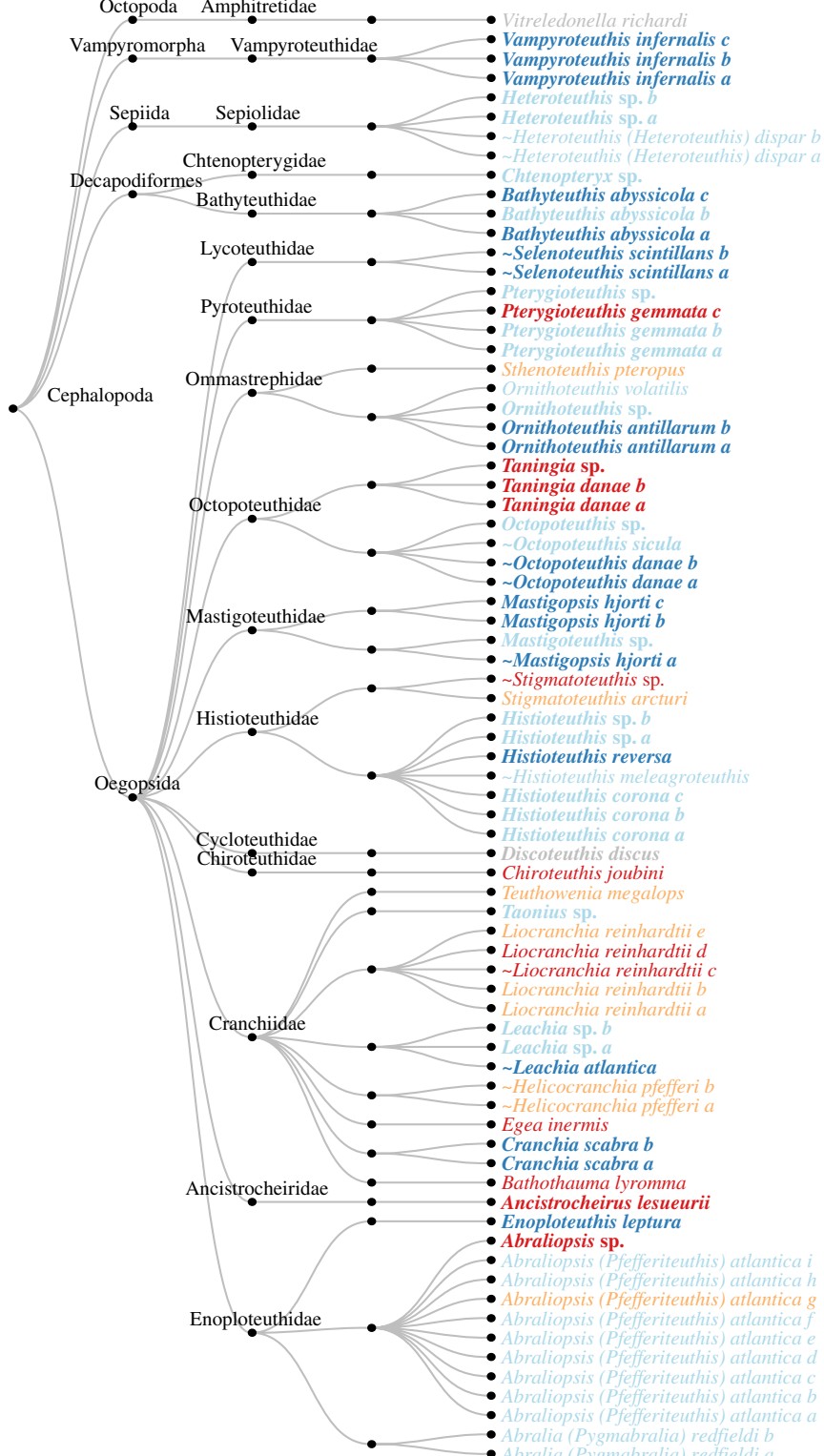

**Figure 4.** BLAST results of sequences of 75 tissue DNA extracts obtained with the *Ceph18S* primer. Morphological ID names in bold indicate that there was a reference sequence to the same taxonomic resolution (genus or species) in GenBank. Grey specimens could not be amplified. Name colours indicate whether the match between morphological and BLAST ID was to species level (dark blue), genus level (light blue), family level (orange) or other taxonomic level (red). A ~ (tilde) indicates there was another BLAST match with the exact same score, i.e. that the taxon could not be unambiguously identified.

*dispar* (the existing partial 18S sequence accession code AF034565 is from a different 18S subregion), *Histioteuthis meleagroteuthis*, *Liocranchia reinhardtii*, *Ornithoteuthis volatilis*, *Sthenoteuthis pteropus*, *Stigmatoteuthis arcturi* and *Teuthowenia megalops*.

Of the 15 amplified species with a reference sequence in GenBank (i.e. excluding the two genus-only taxa and *D. discus*), seven could be unambiguously matched to species level ($B_s = 0.47$): *Bathyteuthis abyssicola*, *Cranchia scabra*, *Enoploteuthis leptura*, *Histioteuthis reversa*, *Mastigopsis hjorti*, *Ornithoteuthis antillarum* and *Vampyroteuthis infernalis*. Some species had an ambiguous BLAST match, i.e. the BLAST match returned multiple species with the same top alignment score including the expected morphologically identified species which, therefore, could not be genetically distinguished from other species. *Leachia atlantica* could not be distinguished from *Leachia lemur*, *Octopoteuthis danae* could not be distinguished from *Octopoteuthis megaptera* and *Selenoteuthis scintillans* could not be distinguished from *Lycoteuthis lorigera*. Species with a GenBank reference sequence that could not be identified were *Ancistrocheirus lesueurii* (best match with Identity = 96, $E$-val = $2.00 \times 10^{-67}$ to *Mastigoteuthis hjorti* which belongs to a different family, as identified in GenBank accession code EU735291 by Lindgren [53], but accepted in WoRMS as *Mastigopsis hjorti* [54]), *Histioteuthis corona* (best match with Identity = 100, $E$-val = $1.00 \times 10^{-63}$ to *Histioteuthis hoylei* as identified in GenBank accession code AY557500 by Lindgren *et al*. [55], but accepted in WoRMS as *Stigmatoteuthis hoylei* [56]), *Octopoteuthis sicula* (best match with Identity = 100, $E$-val = $1.00 \times 10^{-83}$ to *Octopoteuthis danae* and *Octopoteuthis megaptera*), *Pterygioteuthis gemmata* (best match with Identity = 99, $E$-val = $3.00 \times 10^{-47}$ to *Pterygioteuthis microlampas*) and *Taningia danae* (best match with Identity = 93, $E$-val = $2.00 \times 10^{-67}$ to *Lepidoteuthis grimaldii*, both belonging to octopoteuthid families).

Of the 20 amplified genera with a reference sequence at genus level in GenBank, 15 could be identified unambiguously ($B_s = 0.75$). Of the six species that did not have a reference sequence in GenBank at either species or genus level, four could be identified to family level: in the Cranchiidae family (i) *Helicocranchia pfefferi* matched to *Taonius pavo* and *Megalocranchia* sp., (ii) *Liocranchia reinhardtii* matched to *Cranchia scabra*, and (iii) *Teuthowenia megalops* matched to *Taonius pavo*, whereas in the Histioteuthidae family (iv) *Stigmatoteuthis arcturi* matched to *Histioteuthis hoylei* (as identified in GenBank accession code AY557500 by Lindgren *et al*. [55], but accepted in WoRMS as *Stigmatoteuthis hoylei* [56]). Additionally, *Taningia danae* was identified as *Lepidoteuthis grimaldii*, which belong to closely related but different families [57].

Taking into account the availability of reference sequences, a total of 18 of the 30 amplified taxa had the best possible expected outcome (60%). An additional seven taxa could be matched to the next best taxonomic resolution (23%), and three taxa could not be identified to species, genus or family level despite reference sequences being available (10%). A final two taxa could not be identified to species, genus or family level, but also no reference sequences were available, so it is unknown if the lack of references or poor resolution of the primer is responsible (7%).

## 3.3. Comparison to 16S rRNA primer sets

The GenBank nuclear 18S rRNA reference database obtained through Primer-BLAST [48] with *Ceph18S* contained 107 taxa, and was, therefore, similar in size to the SILVA 18S rRNA reference database with 97 taxa. The coverage index and specificity index for *Ceph18S* was similar for this GenBank ($B_c = 0.80$, $B_s = 0.80$) and SILVA database ($B_c = 0.85$, $B_s = 0.78$).

The GenBank mitochondrial 16S rRNA database for both *CephMLS* [41] and *S_Cephalopoda* [42] contained 367 taxa. The coverage index for *CephMLS* and *S_Cephalopoda* was $B_c = 0.82$ and $B_c = 0.72$, respectively, i.e. of similar size and lower than for *Ceph18S*. The specificity index for *CephMLS* and *S_Cephalopoda* was $B_s = 0.69$ and $B_s = 0.46$, i.e. 11–34% lower than for *Ceph18S*. A Venn diagram analysis shows that 95% and 94% of cephalopod taxa can be amplified (figure 5*a*) and identified (figure 5*b*), respectively, by a multi-marker approach with *Ceph18S*, *CephMLS* and *S_Cephalopoda* for eDNA surveys. There is 1% of taxa that can only be amplified by *Ceph18S* and not by *CephMLS* or *S_Cephalopoda*, and 19% of cephalopod taxa can be identified unambiguously by *Ceph18S* but not by *CephMLS* or *S_Cephalopoda*. For comparison, 7% and 0% of taxa can be unambiguously identified only by *CephMLS* or *S_Cephalopoda*, respectively, and 9% can be identified by both 16S primer sets but not by *Ceph18S*. In other words, while the primer sets complement each other only moderately in terms of amplification success, the *Ceph18S* target sequences have a greater taxonomic resolution so that 19% additional taxa can be identified.

# 4. Discussion

A novel universal primer pair named *Ceph18S* targeting the cephalopod nuclear 18S rRNA region was characterized. Due to the relatively small target sequence length of approximately 200 bp, the primer

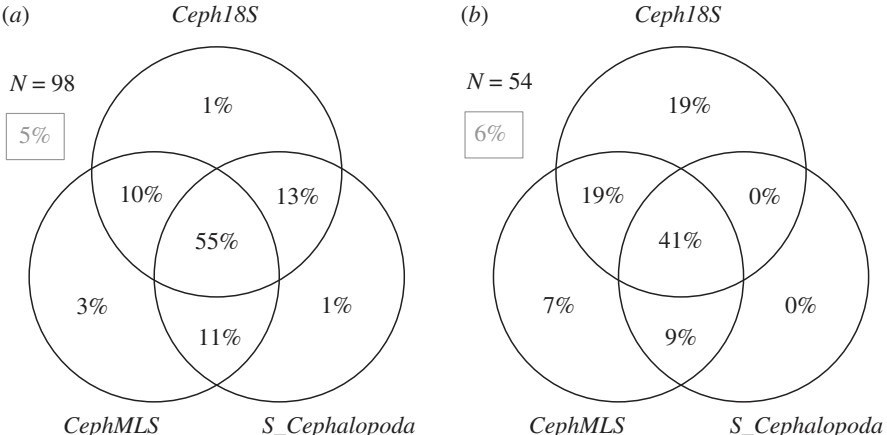

**Figure 5.** Venn diagram of complementarity in (*a*) amplification and (*b*) identification between *Ceph18S*, *CephMLS* [41] and *S_Cephalopoda* [42]. *N* represents (*a*) the number of taxa that were present in both the 16S and 18S GenBank reference databases, and (*b*) the number of taxa amplified by all three primer sets (i.e. the middle plane of (*a*) is the subset used for (*b*)). Grey percentage in the box is the number of taxa that could not be (*a*) amplified and (*b*) identified by either primer set. Sum of percentages may deviate ±1% due to rounding.

pair is suitable for metabarcoding of cephalopod eDNA and can be applied in field studies. We will discuss the *Ceph18S* primer in the context of metabarcoding, where reliable primers are important to avoid false positives (i.e. wrongly identified taxa) and false negatives (i.e. undetected species) [58].

## 4.1. Application of *Ceph18S* in field studies

The *Ceph18S* primer pair targets the V2 variable region of the small-subunit 18S rRNA [52] flanked by relatively conserved regions. This region is suitable for taxonomic assignment [59], with enough variation to allow the identification of a variety of taxa. However, clustering algorithms in metabarcoding pipelines can have difficulties with ribosomal target sequences due to the inherent length variation of rRNA variable regions [60]. As this length variation is present in *Ceph18S* target sequences (131–196 bp), it is recommended to omit clustering and use sequence variants directly, for example, with DADA2 [61], so no clustering threshold has to be chosen and a higher taxonomic resolution can be obtained [62].

According to the coverage index estimated *in silico*, *Ceph18S* should be able to amplify approximately 80–85% of cephalopod species. This coverage index might be slightly underestimated due to missing V2 regions in SILVA reference sequences for some species. Coverage of these species by *Ceph18S* could be checked using tissue DNA extracts if specimens are available. The variation in the annealing sites of the primers indicates that there will be mismatches between the primer and some target species. A small number of mismatches are not always problematic, as amplification might still occur although perhaps suboptimally [63]. This is confirmed by the fact that all but two tested cephalopod DNA extracts (94%) could be amplified with *Ceph18S* empirically. Case in point is the observation that *Vampyroteuthis infernalis* was not amplified *in silico* due to a mismatch between the primer and annealing site but was amplified and correctly identified to species level empirically. The distinctive phylogenetic position of *v. infernalis* in the cephalopod phylogenetic tree [53], and thus its distinctive target sequence, allowed the alignment algorithm to distinguish between *v. infernalis* and target sequences from other taxonomic groups even though the annealing mismatch might potentially have resulted in suboptimal amplification. A possibility to increase the universality of the primer would be to create a degenerate version of *Ceph18S*, i.e. a mixture of very similar primers where one or more nucleotide bases are varied in order to limit mismatches with target species [64].

Overall, the tests indicate that an estimated 44–66% ($B_c \times B_s$) of all cephalopod species can be amplified and correctly identified with the *Ceph18S* primer. In comparison, *S_Cephalopoda* [42] and *CephMLS* [41] are estimated to amplify and identify 33% and 56% of all cephalopod species, respectively. Both the *in silico* and empirical tests indicate that the *Ceph18S* primer pair is not suitable for the detection of octopods, and can give ambiguous results for sepiids, myopsids, octopotheuthids and gonatids. Based on the empirical results, it might appear that the cranchiids are also problematic, but this may also be caused by a severe under-representation of this group in GenBank. If only

specificity to genus level is required, *Ceph18S* performs well with an empirically obtained genus-level specificity of 75%. Furthermore, filtering procedures in the metabarcoding pipeline can increase the number of identified taxa. Any assumptions in these filtering procedures should, however, be considered when interpreting the results. For example, it might be possible to filter based on known biogeographic area, i.e. there are multiple possible species matches to a global reference database, but only one species is known to occur in the studied region and this would omit the detection of, for example, any invasive species.

## 4.2. Reference sequences

The specificity index of *Ceph18S* was 0.78–0.80 as estimated *in silico*, and 0.47 as estimated empirically. However, this estimated empirical specificity index is based on a limited number of identified species ($n = 15$) for which reference sequences were available in GenBank. Of the approximately 700 known cephalopod species, only 88 species are present in the SILVA database and fewer than 300 species have a complete or partial 18S rRNA sequence in GenBank. The reference sequences for cephalopods in GenBank that are currently available are mostly mitochondrial sequences, whereas the 18S rRNA gene is nuclear. For almost half of the taxa that were empirically tested with the *Ceph18S* primer, no 18S rRNA reference sequences were available in GenBank. Our barcoding efforts with *Ceph18S* add new 18S rRNA reference sequences for 13 taxa to GenBank. A comprehensive high-quality reference database is important for metabarcoding primer development and testing because it provides a good overview of nucleotide diversity over the gene and can reduce the occurrence of misidentifications and false negatives [58]. An effort to sequence additional species, as we present here, is, therefore, recommended to aid further primer development and the taxonomic resolution of eDNA metabarcoding surveys [65]. If a proper reference library is not available, alternative biodiversity estimates based on sequence diversity (i.e. omitting direct taxonomic assignment) could be used [66].

There are five species for which the *Ceph18S* target sequences did not match to the expected species in GenBank, even though a representative reference sequence was available.

A first explanation could be a wrong morphological identification assigned to the DNA sequence. For example, the taxa in the Histioteuthidae family are relatively difficult to distinguish, which may have caused a misidentification of our *Histioteuthis corona* or its matching GenBank sequence *Histioteuthis hoylei*. However, we deem this explanation unlikely, as all morphological identifications in both this paper and for the GenBank reference sequences were done by cephalopod experts (H.-J.T.H. and Annie Lindgren, respectively).

A second explanation could be the existence of cryptic species, where species are morphologically similar but genetically different. Although widespread existence of cryptic oceanic species has been suggested [67] and has been shown for some cephalopod taxa [68,69], no cryptic species complexes have been reported for the species with GenBank mismatches. Additionally, the taxonomy of the Octopoteuthidae is problematic with evidence of genetic similarity between *Octopoteuthis sicula*, *O. danae* and *O. megaptera*, which does not support the distinction of multiple species [70] and explains our 100% match of *O. sicula* to *O. danae* and *O. megaptera* with our relatively short *Ceph18S* target sequence.

A third explanation for the mismatches is that the relatively short target sequence length of *Ceph18S* in some cases cannot provide enough resolution to account for natural variability for a reliable identification, especially if the species is under-represented in GenBank. Three of the five mismatched species did match to the correct genus. Target sequences within a taxon can be expected to be relatively similar, so that a couple of different nucleotide bases, either due to natural variability or erroneous base calls in the sequencing process, can induce mismatches especially in short target sequences. The remaining two mismatched species with hits outside the expected genus had low identities to their best match (93%, 96%) and only one representative reference sequence available in GenBank. The quality of all our barcoded sequences was reviewed and approved, and repeated sequencing of the same individuals gave consistent results. For example, the same specimen of *Taningia danae*, which was reliably identified morphologically, was sequenced twice with consistent target sequences and closest match of 93% to *Lepidoteuthis grimaldii*. Therefore, it is likely this sequence of *T. danae* reflects natural variability in this partial 18S rRNA region for the species.

## 4.3. Biodiversity surveys with a multi-marker approach

Extrapolating the *Ceph18S* $B_c \times B_s$ value to the known approximately 700 cephalopod species, it is estimated about 310–460 species can be amplified and identified with *Ceph18S*. Additionally, 19% of

taxa that are amplified by all three primers can only be unambiguously identified by *Ceph18S* and 16% can only be unambiguously identified by the 16S primer sets. Therefore, a multi-marker approach, where multiple universal cephalopod primers are combined, can increase overall coverage and specificity when applying eDNA analysis for cephalopod biodiversity studies [29]. Note that the 16S rRNA primer set *S-Cephalopoda* from [42] targets a subregion of the target sequence amplified by the 16S primer set *CephMLS* from [41], which are, therefore, similar in their resolution.

Employing a multi-marker approach has several advantages. Firstly, the highest confidence for species detection is obtained when a species is found in multiple samples and by multiple markers [71]. However, solely applying such a stringent definition would omit species that can be detected by one marker only. In the complementary case of 18S and 16S cephalopod primer sets, 35% of cephalopod species would not be detected with confidence under this stringent definition as they are only identified by either one, and not by both. Therefore, we recommend appreciating the complementary nature of primers sets and assigning a confidence value to a species detection rather than fully discarding single marker detections. Secondly, employing multiple markers is useful with limited availability of reference sequences [72]. By searching for multiple markers, the reference database can include species that might have a reference sequence for one region, but not the other. To successfully employ the multi-marker approach, it is important to understand the limitations of each marker. For example, markers may have different affinities to different taxonomic groups [32] and there might be differences in detectability between nuclear and mitochondrial eDNA [73–75]. Studies into marker specificity and limitations, like the current study, can, therefore, help interpret eDNA metabarcoding results.

If the specificity of certain cephalopod groups is still low even with the combined marker approach, additional universal primer pairs can be developed (e.g. based on the 28S region, [55]), potentially targeting only a subgroup within the cephalopods. Additionally, the usage of slightly varying primer annealing temperatures, like in a touch-down PCR [76], can reveal more taxa than using a single annealing temperature. High temperatures favour rare but perfectly matching sequences, but lower temperatures generally recover broader diversity as it allows annealing mismatches, and taxa found at different temperatures are not strictly subsets of each other and thus add to overall richness [77].

Even though primer sets for specific cephalopod taxa allow identification of specific elusive species [25] and might complement a universal cephalopod primer for problematic taxa, it would be difficult to do cephalopod biodiversity assays using only species-specific primers. For example, regional reported species diversity is 32 in the Arctic [78], 54 in the Antarctic [78], 68 at the Southwest Indian Ocean Ridge [79], 70 around the Kermadec Islands [80], 77 near Bear Seamount [81], up to 85 around the Canary Islands [82]. Additionally, such an approach would require pre-existing knowledge of local cephalopod diversity, which is often lacking. Therefore, an eDNA metabarcoding approach with multiple markers of which the strengths and limitations are tested is a suitable complementary method for further studies into local cephalopod biodiversity patterns.

# 5. Conclusion

Despite the pivotal role of cephalopods in marine food webs, knowledge on their diversity and distribution still has major gaps, especially in oceanic regions. Metabarcoding of eDNA is a proven powerful technique for biodiversity surveys, but it has not yet been applied to the study of these elusive organisms. This study characterized a new universal metabarcoding primer pair for the analysis of cephalopod eDNA targeting the nuclear 18S rRNA gene in order to complement published mitochondrial 16S rRNA primer sets [41,42]. The developed *Ceph18S* primer pair amplifies an approximately 200 bp target sequence estimated to be able to identify about 310–460 cephalopod species. *Ceph18S* is estimated to amplify and identify 8–31% more cephalopod species than the 16S rRNA primer sets. Furthermore, 19% of taxa amplified by both the 16S and 18S rRNA primer sets can only be identified with *Ceph18S*, thereby increasing overall taxonomic resolution in a multi-marker metabarcoding approach. *Ceph18S* is not suitable for the detection of octopods, and should be used with caution on sepiids, myopsids, octopotheuthids and gonatids. The submitted barcodes to GenBank add new 18S rRNA partial sequences for 13 cephalopod taxa previously absent in the GenBank database.

The *Ceph18S* primer pair is currently being applied in metabarcoding studies of cephalopod eDNA from epi-, meso- and bathypelagic depths [83]. The preliminary results of this study show that *Ceph18S* can successfully be applied in a metabarcoding workflow on a large dataset and can successfully amplify and identify cephalopod eDNA. Therefore, the *Ceph18S* primer pair, potentially in degenerate form or combined with other universal cephalopod primers, is a useful new molecular tool that can be

used alongside other sampling methods for studying cephalopod diversity and distribution from shallow, coastal waters to the pelagic and deep-sea environments.

Ethics. Cabo Verde has not ratified the Nagoya protocol. To fulfil the national ABS regulations of Cabo Verde, we obtained the required permit for the publication of results based on samples collected in Cabo Verde waters from the Direccao Nacional do Ambiente (National Directorate for the Environment of Cabo Verdes). Permission for fieldwork and publication of results was granted by Ministério da Agricultura e Ambiente of Cape Verde and Agência Marítima e Portuária of Cape Verde.

Data accessibility. All barcoded sequences are accessible via GenBank accession numbers MT680727–MT680790 and MT680792–MT680800. The electronic supplementary material includes the Supplementary Data and Code zip-file, which contains the code and data used for data analysis and the creation of figures 1, 3 and 4, and Supplementary Table S1, which contains sample station information.

Authors' contributions. D.S.W.d.J. carried out the molecular laboratory work, performed the bioinformatics and data analysis and drafted the manuscript; V.M. participated in the design of the study, collected field data, helped with the laboratory work and participated in data analysis. T.B. helped with the laboratory work and data analysis. O.P. participated in conceiving and designing the study. T.B.H.R. participated in conceiving and designing the study. H.-J.T.H. conceived and designed the study, collected field data, performed morphologies species identifications and coordinated the study. All authors critically revised the manuscript, gave final approval for publication and agree to be held accountable for the work performed therein.

Competing interests. We declare we have no competing interests.

Funding. This research is funded by the Deutsche Forschungsgemeinschaft (DFG) under grant HO 5569/2-1 (Emmy Noether Junior Research Group) awarded to H.-J.T.H., and by GEOMAR's Programme Oriented Funding III OCEANS programme.

Acknowledgements. We thank the Institute of Clinical Molecular Biology in Kiel for providing Sanger sequencing as supported in part by the DFG Clusters of Excellence 'Precision Medicine in Chronic Inflammation' and 'ROOTS'. We thank T. Naujoks, Dr D. Langfeldt and Dr B. Löscher for technical support. We thank Dr Heino Fock (chief scientist of WH383) and Dr Stephanie Czudaj for the collaboration and opportunity to collect cephalopods in the eastern Atlantic which were used in this study.

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
