## [Peer Review File · Royal Society Open Science]

Review History

RSOS-201388.R0 (Original submission)

Review form: Reviewer 1

Is the manuscript scientifically sound in its present form?

Yes

Are the interpretations and conclusions justified by the results?

Yes

Is the language acceptable?

Yes

Do you have any ethical concerns with this paper?

No

Have you any concerns about statistical analyses in this paper?

No

Recommendation?

Accept with minor revision (please list in comments)

Comments to the Author(s)

The authors of "A novel metabarcoding primer pair for environmental DNA analysis of Cephalopoda (Mollusca) targeting the nuclear 18S rRNA region" have developed a new eDNA metabarcoding primers set to target Cephalopoda taxa within water samples. This is a useful primers set that has the potential to advance future Cephalopoda assessments through eDNA metabarcoding. The paper is well written, and the methods and results presented are informative with implications relevant to the research topic. I was very satisfied to read such a well-organized and descriptive study and strongly recommend it for publication in the journal after some minor modifications (see Appendix A). These modifications are mostly related to providing further information on primer testing and further development within the discussion. Once these additions are added, this publication will be a valuable addition to the journal and a useful source for future Cephalopoda monitoring efforts, and the growing literature of targeted metabarcoding eDNA studies.

Review form: Reviewer 2**Is the manuscript scientifically sound in its present form?**

Yes

Are the interpretations and conclusions justified by the results?

Yes

Is the language acceptable?

Yes

Do you have any ethical concerns with this paper?

No

Have you any concerns about statistical analyses in this paper?

No

Recommendation?

Accept with minor revision (please list in comments)

Comments to the Author(s)

This is an interesting manuscript that contributes a new primer set to analyze for cephalopods in eDNA samples. While there are still many unknowns involving eDNA processes, this primer set will be helpful as it will allow for cephalopods to be included in the analyses. I have only a few comments/edits here to address:

1. In text citations: should these be in chronological order? Most journals require that, please check this journal to align with their guidelines.
2. Lines 81-86: comment: there needs to be extreme caution with publications stating new ranges, etc using eDNA methods. Just my opinion as I read this section....
3. Comment: I think it strengthens your paper by having the empirical testing piece. There are many, many incorrect sequences found in Genbank which could skew your findings. (relating to lines 265-305).

Lines 50-52: Oddly worded sentence, please reword to clarify this statement.

Line 85: delete not yet and replace with none specifically

Line 307: of the 107 in Genbank- were they all reliably identified? Many readers might be skeptical of the use of Genbank but I absolutely understand the need to use it as it's the best ref we have currently for these types of studies.

Conclusions: It is great that additional, reliable sequences are being added to Genbank. I think there is still quite a way to go before eDNA studies are reliable but for using this primer set as part of the tools to study biodiversity and locations of cephalopods will be an aid to the field.

Decision letter (RSOS-201388.R0)

Dear Mrs de Jonge,

On behalf of the Editors, we are pleased to inform you that your Manuscript RSOS-201388 "A novel metabarcoding primer pair for environmental DNA analysis of Cephalopoda (Mollusca) targeting the nuclear 18S rRNA region" has been accepted for publication in Royal Society Open Science subject to minor revision in accordance with the referees' reports. Please find the referees' comments along with any feedback from the Editors below my signature.

Please submit your revised manuscript and required files (see below) no later than 7 days from today's (ie 08-Dec-2020) date. Note: the ScholarOne system will 'lock' if submission of the revision is attempted 7 or more days after the deadline. If you do not think you will be able to meet this deadline please contact the editorial office immediately.

on behalf of the Associate Editor, and Professor Pete Smith (Subject Editor)

Associate Editor Comments to Author:

Thank you for your patience while the journal found reviewers for your work - we regret that delays have been more common this year, but are grateful for your support. The reviewers recommend the paper may be accepted once you have completed a number of revisions - please ensure you fully address these changes.

Reviewer comments to Author:

Reviewer: 1

Comments to the Author(s)

The authors of "A novel metabarcoding primer pair for environmental DNA analysis of Cephalopoda (Mollusca) targeting the nuclear 18S rRNA region" have developed a new eDNA metabarcoding primers set to target Cephalopoda taxa within water samples. This is a useful primers set that has the potential to advance future Cephalopoda assessments through eDNA metabarcoding. The paper is well written, and the methods and results presented are informative with implications relevant to the research topic. I was very satisfied to read such a well-organized and descriptive study and strongly recommend it for publication in the journal after some minor modifications. These modifications are mostly related to providing further information on primer testing and further development within the discussion. Once these additions are added, this publication will be a valuable addition to the journal and a useful source for future Cephalopoda monitoring efforts, and the growing literature of targeted metabarcoding eDNA studies.

Reviewer: 2

Comments to the Author(s)

This is an interesting manuscript that contributes a new primer set to analyze for cephalopods in eDNA samples. While there are still many unknowns involving eDNA processes, this primer set will be helpful as it will allow for cephalopods to be included in the analyses. I have only a few comments/edits here to address:

1. In text citations: should these be in chronological order? Most journals require that, please check this journal to align with their guidelines.
2. Lines 81-86: comment: there needs to be extreme caution with publications stating new ranges, etc using eDNA methods. Just my opinion as I read this section...
3. Comment: I think it strengthens your paper by having the empirical testing piece. There are many, many incorrect sequences found in Genbank which could skew your findings. (relating to lines 265-305).

Lines 50-52: Oddly worded sentence, please reword to clarify this statement.

Line 85: delete not yet and replace with none specifically

Line 307: of the 107 in Genbank- were they all reliably identified? Many readers might be skeptical of the use of Genbank but I absolutely understand the need to use it as it's the best ref we have currently for these types of studies.

Conclusions: It is great that additional, reliable sequences are being added to Genbank. I think there is still quite a way to go before eDNA studies are reliable but for using this primer set as part of the tools to study biodiversity and locations of cephalopods will be an aid to the field.

===PREPARING YOUR MANUSCRIPT===

===PREPARING YOUR REVISION IN SCHOLARONE===

- An individual file of each figure (EPS or print-quality PDF preferred [either format should be produced directly from original creation package], or original software format).
 - An editable file of each table (.doc, .docx, .xls, .xlsx, or .csv).
 - An editable file of all figure and table captions.
- Note: you may upload the figure, table, and caption files in a single Zip folder.
- Any electronic supplementary material (ESM).
 - If you are requesting a discretionary waiver for the article processing charge, the waiver form must be included at this step.
 - If you are providing image files for potential cover images, please upload these at this step, and inform the editorial office you have done so. You must hold the copyright to any image provided.
 - A copy of your point-by-point response to referees and Editors. This will expedite the preparation of your proof.

- Ensure that your data access statement meets the requirements at <https://royalsociety.org/journals/authors/author-guidelines/#data>. You should ensure that you cite the dataset in your reference list. If you have deposited data etc in the Dryad repository, please only include the 'For publication' link at this stage. You should remove the 'For review' link.
- If you are requesting an article processing charge waiver, you must select the relevant waiver option (if requesting a discretionary waiver, the form should have been uploaded at Step 3 'File upload' above).
- If you have uploaded ESM files, please ensure you follow the guidance at <https://royalsociety.org/journals/authors/author-guidelines/#supplementary-material> to include a suitable title and informative caption. An example of appropriate titling and captioning may be found at https://figshare.com/articles/Table_S2_from_Is_there_a_trade-off_between_peak_performance_and_performance_breadth_across_temperatures_for_aerobic_scops_in_teleost_fishes_/3843624.

Author's Response to Decision Letter for (RSOS-201388.R0)

See Appendix B.

Decision letter (RSOS-201388.R1)

Dear Mrs de Jonge,

It is a pleasure to accept your manuscript entitled "A novel metabarcoding primer pair for environmental DNA analysis of Cephalopoda (Mollusca) targeting the nuclear 18S rRNA region" in its current form for publication in Royal Society Open Science.

on behalf of Professor Pete Smith (Subject Editor)
openscience@royalsociety.org

Appendix A

A novel metabarcoding primer pair for environmental DNA analysis of Cephalopoda (Mollusca) targeting the nuclear 18S rRNA region

The authors of “A novel metabarcoding primer pair for environmental DNA analysis of Cephalopoda (Mollusca) targeting the nuclear 18S rRNA region” have developed a new eDNA metabarcoding primer set to target Cephalopoda taxa within water samples. This is a useful primer set that has the potential to advance future Cephalopoda assessments through eDNA metabarcoding. The paper is well written, and the methods and results presented are informative with implications relevant to the research topic. I was very satisfied to read such a well-organized and descriptive study and strongly recommend it for publication in the journal after some minor modifications. These modifications are mostly related to providing further information on primer testing and further development within the discussion. Once these additions are added, this publication will be a valuable addition to the journal and a useful source for future Cephalopoda monitoring efforts, and the growing literature of targeted metabarcoding eDNA studies.

Introduction:

Line 61-63: Citation of Venter et al. 2004 refers to a microbiology environmental DNA based study, which greatly differs from the current fast rise and interest in Metazoan detection with eDNA. I suggest cited Ficetola et al. 2008. “Species detection using environmental DNA from water samples”.

Line 72-74: You should be able to find ample citations for eDNA used in the Marine environment, rather than citing Ficetola et al. 2008 here.

Line 75-77: Write out which group Djurhuus et al., 2018 targeted with marine eDNA samples.

Line 78: “Focussed” is spelled wrong.

Line 80: Change “they already provided” to “they were able to provide”. “They already provided” makes it sound like the research into cephalopod distribution is completed.

Line 87-91: It is also important to note that “universal primers” should not amplify non-target taxa.

Line 101-104: Likewise here, you can state the previous 16S and Cytb primers are cephalopod specific. To the casual reader, universal primers might be confusing, since you are discussing targeting cephalopods.

Methods:

Lines 122-124: How long was the 18S region from the Silva database?

Lines 164-165: “Another cephalopod 18S database in addition to the SILVA database was generated in this same manner from GenBank”. This line is confusing to me, because you already detailed the 18S Genbank database in the Reference Database section. So I am confused in what this is referring to.

Lines 178-179: What tissue was used for DNA extraction? Was the entire specimen placed into EtOH, or just a piece of tissue?

Line 194-195: I am assuming the temperature gradient increased by a total of 3°C, but this is not clear. It reads as if each of the 5 steps increases by 3°C.

Lines 197-199: What other species were the primersets tested on?

Lines 220-221: What software did you use to do this?

Results:

Lines 261-264: This is an interesting finding, and you could verify this claim by examining the primer region of these sequences against your primers.

Lines 270-271: “although sometimes with suboptimal amplicon quality” What does this mean exactly? Were these amplified DNA that smeared in gels?

Also, for the taxa that did not amplify, did you test a universal primerset (such as Folmer et al. (COI)) on the DNA extraction. To verify the extracted DNA was of amplifiable quality and that of the cephalopod species of interest?

Lines 280-290: This section is a bit confusing. So these species have representative sequences within Genbank, however their best BLAST match was to a different species?

Lines 320-322: This belongs in the discussion and not the results

Discussion:

Lines 356-358: This is important information that should go into the results in the Database description section.

Lines 362-367: This statement should be in the results section.

Lines 379-380: Can you add a column to a table that points out which species belong to these groups. That will make it easy for a reader to visualize this point.

Lines 391-393: I am also curious how many unique taxa can only be identified by the 16S primersets.

Additions to the Discussion:

Examples of Cephalopoda regional diversity

As a novice in understanding localized cephalopod diversity, I wonder if a better approach would be to use many species-specific qPCR assays, rather than an 18S metabarcoding approach. Therefore, providing some information on cephalopod biodiversity hotspots and the number of species in a localized region, would provide the reader with the need for a metabarcoding primerset.

Previous eDNA studies have analyzed metabarcoding data with a multi-marker approach to improve species detection (such as Evans et al. 2017, Li et al. 2018), and I think you need to add additional information about these studies when discussing a mutlimarker cephalopod approach with 18S and 16S.

Evans, N. T., Li, Y., Renshaw, M. A., Olds, B. P., Deiner, K., Turner, C. R., ... & Pfrender, M. E. (2017). Fish community assessment with eDNA metabarcoding: effects of sampling design and bioinformatic filtering. *Canadian Journal of Fisheries and Aquatic Sciences*, 74(9), 1362-1374.

Li, Y., Evans, N. T., Renshaw, M. A., Jerde, C. L., Olds, B. P., Shogren, A. J., ... & Pfrender, M. E. (2018). Estimating fish alpha-and beta-diversity along a small stream with environmental DNA metabarcoding. *Metabarcoding and Metagenomics*, 2, e24262.

Furthermore, with discussion about a multimarker 18S and 16S combination, I think it is imperative that you mention possible discrepancies in detectability between nuclear and mitochondrial eDNA.

Bylemans, J. et al. Does size matter? An experimental evaluation of the relative abundance and decay rates of aquatic environmental DNA. *Environ. Sci. Technol.* 52, 6408–6416 (2018).
<https://doi.org/10.1021/acs.est.8b01071>

Jo, T. et al. Estimating shedding and decay rates of environmental nuclear DNA with relation to water temperature and biomass. *Environ. DNA*, 2, 140–151 (2020).
<https://doi.org/10.1002/edn3.51>

Moushomi, R. et al. Environmental DNA size sorting and degradation experiment indicates the state of *Daphnia magna* mitochondrial and nuclear eDNA is subcellular. *Sci. Rep.* 9, 1–9. (2019).
<https://doi.org/10.1038/s41598-019-48984-7>

Figures: I do not see any figure legends.

Appendix B

General response: The feedback by the reviewers was much appreciated and used to improve the manuscript. Below is a detailed response to each individual comment. We look forward to your evaluation.

Reviewer: 1

Comments to the Author(s)

The authors of “A novel metabarcoding primer pair for environmental DNA analysis of Cephalopoda (Mollusca) targeting the nuclear 18S rRNA region” have developed a new eDNA metabarcoding primerset to target Cephalopoda taxa within water samples. This is a useful primerset that has the potential to advance future Cephalopoda assessments through eDNA metabarcoding. The paper is well written, and the methods and results presented are informative with implications relevant to the research topic. I was very satisfied to read such a well-organized and descriptive study and strongly recommend it for publication in the journal after some minor modifications. These modifications are mostly related to providing further information on primer testing and further development within the discussion. Once these additions are added, this publication will be a valuable addition to the journal and a useful source for future Cephalopoda monitoring efforts, and the growing literature of targeted metabarcoding eDNA studies.

Introduction:

Line 61-63: Citation of Venter et al. 2004 refers to a microbiology environmental DNA based study, which greatly differs from the current fast rise and interest in Metazoan detection with eDNA. I suggest cited Ficetola et al. 2008. “Species detection using environmental DNA from water samples”.

Our response: We have changed this citation to Ficetola et al. (2008) and moved the citation of Venter et al. (2004) to the statement about the origin of metabarcoding.

Line 72-74: You should be able to find ample citations for eDNA used in the Marine environment, rather than citing Ficetola et al. 2008 here.

Our response: We have now included a couple of recent examples where eDNA analysis has been used for elusive species and/or in remote areas. We have nevertheless kept the reference to Ficetola et al. (2008) as they specifically recommend this method for such cases studies.

Line 75-77: Write out which group Djurhuus et al., 2018 targeted with marine eDNA samples.

Our response: Changed to: “Metabarcoding of eDNA from seawater has mostly been used to identify fishes (Thomsen et al., 2012, Andruszkiewicz et al., 2017; Sigsgaard et al., 2017), assess overall (metazoan) eukaryotic diversity (Stat et al. 2017, Djurhuus et al., 2018, Günther et al. 2018, Stefanni et al. 2018,), but to our knowledge has not been used to focus on specific taxonomic groups like cephalopods.”

Line 78: “Focussed” is spelled wrong.

Our response: Changed to “focused”.

Line 80: Change “they already provided” to “they were able to provide”. “They already provided” makes it sound like the research into cephalopod distribution is completed.

Our response: Changed as suggested.

Line 87-91: It is also important to note that “universal primers” should not amplify non-target taxa.

Our response: Changed to: “Ideally, a pair of universal primers will target the largest possible taxonomic group of interest, unambiguously identify all species, while not amplifying non-target taxa.”

Line 101-104: Likewise here, you can state the previous 16S and Cytb primers are cephalopod specific. To the casual reader, universal primers might be confusing, since you are discussing targeting cephalopods.

Our response: Changed to “So far, two sets of universal primers specifically targeting cephalopods have been published, ...”

Methods:

Lines 122-124: How long was the 18S region from the Silva database?

Our response: The 18S region from the SILVA database ranged from 423 to 2610 bp. This number is now included in the text as: “... and included 146 sequences from 88 species ranging from 423 to 2610 bp”.

Lines 164-165: “Another cephalopod 18S database in addition to the SILVA database was generated in this same manner from GenBank”. This line is confusing to me, because you already detailed the 18S Genbank database in the Reference Database section. So I am confused in what this is referring to.

Our response: We understand the confusion, and tried to be more explicit in the text about this choice.

Methods:

“The 18S primer development process was based on two reference databases: one from SILVA with 146 sequences, and one from GenBank with 31 sequences. The latter has significantly less sequences than the SILVA database, caused by our specific filtering choices to avoid non-overlapping sequences which would have obstructed the development process. We calculated the B_c and B_s indices for our new primer set from the SILVA database during the development process. However, we felt that a comparison between these SILVA derived 18S indices and the Primer-Blast derived 16S indices would be biased. SILVA is specific about which GenBank sequences are admitted into the alignment, and some GenBank sequences might have been left out. To ensure an unbiased comparison between 16S and 18S coverage and specificity indices, we obtained a third 18S database by using the newly developed 18S primer sequence in a GenBank Primer-BLAST (Jian et al., 2012). This third GenBank database could not have been obtained at the start when we had not yet developed the primer set.”

Results:

“The GenBank nuclear 18S rRNA reference database obtained through Primer-BLAST (Jian et al., 2012) with *Ceph18S* contained 107 taxa, and was therefore similar in size to the SILVA 18S rRNA reference database with 97 taxa. The coverage index and specificity index for *Ceph18S* was similar for this GenBank ($B_c = 0.80$, $B_s = 0.80$) and SILVA database ($B_c = 0.85$, $B_s = 0.78$).”

Lines 178-179: What tissue was used for DNA extraction? Was the entire specimen placed into EtOH, or just a piece of tissue?

Our response: Changed to “The specimens were morphologically identified by HJH, and the full specimen (for small individuals) or a part of an arm (for larger individuals) was stored in a 2 ml tube with ethanol.”

Line 194-195: I am assuming the temperature gradient increased by a total of 3°C, but this is not clear. It reads as if each of the 5 steps increases by 3°C.

Our response: It is indeed 5 steps of 3°C, not a full gradient of 3°C. We explained this in the text as follows:

Methods:

“The temperature gradient started at 3°C below the lowest T_m of the primer set and increased with five steps of 3°C each. This temperature gradient was chosen to account for the expected increase in optimal annealing temperature due to the KAPA Hifi kit, and a decrease in optimal annealing temperature due to the DMSO in the PCR mix, which together could cause deviation from the theoretical optimal annealing temperature by several degrees Celsius.”

Results:

“The optimal annealing temperature for *Ceph18S* in the PCR master mix used in this study was found to be 62°C. This differs from the calculated T_m (Table 2) as both the KAPA reagents and DMSO in our PCR master mix alter the annealing temperature.”

Lines 197-199: What other species were the primersets tested on?

Our response: As we tested tissue from a rather long list of species, we want to refrain from including the complete list in the Methods section text. Instead, we changed the sentence to: “...the same PCR procedure was conducted on more cephalopod tissue DNA extracts (30 species, Fig. 4, Table S1), ...”. In the results section we also provide more detail on the larger set of tested specimens.

Lines 220-221: What software did you use to do this?

Our response: Added to text: “Low-quality ends and primers were trimmed manually from the Sanger sequences, which were then manually checked and edited using 4Peaks V1.8 (Griekspoor & Groothuis, 2004), and subsequently assembled using AliView V1.24 (Larsson, 2014).”

Results:

Lines 261-264: This is an interesting finding, and you could verify this claim by examining the primer region of these sequences against your primers.

Our response: As suggested by the reviewer we examined these results more closely to explain this finding. Closer inspection revealed incomplete SILVA reference sequences, i.e. lacking a V2 region around which the primer set anneals. Upon this finding also the reference sequences for unamplified taxa *in silico* were checked for missing V2 regions. This might have caused an underestimation for the coverage index, which could be checked by amplifying tissue extracts of these taxa. Unfortunately, we did not have access to specimens to empirically test coverage of *Ceph18S* of these taxa. This is now addressed in the text as:
Results:

“Taxa for which some reference sequences were amplified but not all, were *Chtenopteryx sicula*, *Loligo forbesi*, *Sepia elegans*, *Sepiella inermis*, and *Todaropsis eblanae*. Further inspection revealed that some reference sequences of these species were incomplete i.e. omitting at least the V2 region around which the *Ceph18S* primer set anneals. Taxa that were not amplified at all due to a lacking reference V2 region were *Eledone cirrhosa*, *Euprymna scolopes*, *Hapalochlaena maculosa*, *Loligo vulgaris*, *Octopus vulgaris*, *Opisthoteuthis* sp., and *Rossia macrosoma*. Taxa that were not amplified even though a reference V2 region was available were *Alloteuthis* sp., *Bathypolypus* sp., *Cirrothauma murrayi*, *Pyroteuthis margaritifera*, *Sepia pharaonis*, *Sepioloidea lineolata*, *Spirula spirula*, and *Vampyroteuthis infernalis*.”

Discussion:

“According to the coverage index estimated *in silico*, *Ceph18S* should be able to amplify ~80-85% of cephalopod species. This coverage index might be slightly underestimated due to missing V2 regions in SILVA reference sequences for some species. Coverage of these species by *Ceph18S* could be checked using tissue DNA extracts if specimens are available.”

Lines 270-271: “although sometimes with suboptimal amplicon quality” What does this mean exactly? Were these amplified DNA that smeared in gels? Also, for the taxa that did not amplify, did you test a universal primerset (such as Folmer et al. (COI)) on the DNA extraction to verify the extracted DNA was of amplifiable quality and that of the cephalopod species of interest?

Our response: Changed to ‘suboptimal sequence quality’, as we refer to some sequences with usable though weaker base-calling signals in the chromatogram. We did not use another primer set, like Folmer’s COI, to verify the quality of the samples without amplification. We agree with the reviewer this would have strengthened the results. However, we always checked the concentration of the DNA extracts, and wherever possible, we did extract more DNA from unamplified species and retried the *Ceph18S* primer in case poor DNA sample quality would have caused the lack of amplification. As can be seen from Figure 4, on multiple occasions we tested the *Ceph18S* on the same species more than once, with fairly consistent results. For our addition to the text about this feedback, please see response to ‘Lines 280-290’.

Lines 280-290: This section is a bit confusing. So these species have representative sequences within Genbank, however their best BLAST match was to a different species?

Our response: Yes, some *Ceph18S* sequences did not match to the expected species in GenBank even with a reference sequence available. There are various explanations for this,

which we now discuss in the manuscript (see below). In summary, these mismatches might occur due to misidentification, cryptic species, problematic taxonomy, and a combination of low-resolution of the short target sequence and underrepresentation of species in GenBank. Upon revision we decided one morphological identification could not be 100% confirmed (*Ommastrephes bartramii*), which is therefore removed from the analysis, and all relevant values are adjusted accordingly.

Discussion:

“

There are five species for which the *Ceph18S* target sequences did not match to the expected species in GenBank, even though a representative reference sequence was available.

A first explanation could be a wrong morphological identification assigned to the DNA sequence. For example, the taxa in the Histioteuthidae family are relatively difficult to distinguish, which may have caused a misidentification of our *Histioteuthis corona* or its matching GenBank sequence *Histioteuthis hoylei*. However, we deem this explanation unlikely, as all morphological identifications in both this paper and for the GenBank reference sequences were done by cephalopod experts (HJH and Annie Lindgren respectively).

A second explanation could be the existence of cryptic species, where species are morphologically similar but genetically different. Although wide-spread existence of cryptic oceanic species has been suggested (66) and has been shown for some cephalopod taxa (67,68), no cryptic species complexes have been reported for the species with GenBank mismatches. Additionally, the taxonomy of the Octopoteuthidae is problematic with evidence of genetic similarity between *Octopoteuthis sicula*, *O. danae*, and *O. megaptera*, which does not support the distinction of multiple species (69) and explains our 100% match of *O. sicula* to *O. danae* and *O. megaptera* with our relative short *Ceph18S* target sequence.

A third explanation for the mismatches is that the relatively short target sequence length of *Ceph18S* in some cases cannot provide enough resolution to account for natural variability for a reliable identification, especially if the species is underrepresented in GenBank. Three of the five mismatched species did match to the correct genus. Target sequences within a taxon can be expected to be relatively similar, so that a couple of different nucleotide bases, either due to natural variability or erroneous base calls in the sequencing process, can induce mismatches especially in short target sequences. The remaining two mismatched species with hits outside the expected genus had low identities to their best match (93%, 96%) and only one representative reference sequence available in GenBank. The quality of all our barcoded sequences was reviewed and approved, and repeated sequencing of the same individuals gave consistent results. For example, the same specimen of *Taningia danae*, which was reliably identified morphologically, was sequenced twice with consistent target sequences and closest match of 93% to *Lepidoteuthis grimaldii*. Therefore, it is likely this sequence of *T. danae* reflects natural variability in this partial 18S rRNA region for the species.

”

Lines 320-322: This belongs in the discussion and not the results

Our response: We would like to keep a sentence here highlighting this result. However, we recognize there is some speculation in this sentence about variability within the target sequence that would better suit the discussion. Hence, we have adjusted the sentence to “In other words, while the primer sets complement each other only moderately in terms of amplification success, the *Ceph18S* target sequences have a greater taxonomic resolution so that 19% additional taxa can be identified”.

Discussion:

Lines 356-358: This is important information that should go into the results in the Database description section.

Our response: This information is already present in the referred result section ‘3.2. Ceph18S resolution’ in the sentence: “Of the 15 amplified species with a reference sequence in GenBank (i.e. excluding the two genus-only taxa and *D. discus*), seven could be unambiguously matched to species level ($B_s = 0.47$)”. In the discussion we merely highlight this result.

Lines 362-367: This statement should be in the results section.

Our response: We have moved this statement to the results section, and now only shortly highlight this in the Discussion.

Lines 379-380: Can you add a column to a table that points out which species belong to these groups. That will make it easy for a reader to visualize this point.

Our response: We thank the reviewer for this suggestion. However, figures 3 and 4 show all tested taxa arranged by their respective taxonomic groups. The nodes of the graphs are annotated with both the order and family names, and include all groups mentioned in the text. We think these figures can therefore better help visualize this point than a table. It would be possible to add further classification to Supplementary Table 1, but then it would become a very crowded table in our opinion.

Lines 391-393: I am also curious how many unique taxa can only be identified by the 16S primersets.

Our response: This information can be found in Figure 5B, and is now also included in the results section:

“For comparison, 7% and 0% of taxa can be unambiguously identified only by *CephMLS* or *S_Cephalopoda* respectively, and 9% can be identified by both 16S primer sets but not by *Ceph18S*.”

And in the discussion section:

“Additionally, 19% of taxa that are amplified by all three primers can only be unambiguously identified by *Ceph18S* and 16% can only be unambiguously identified by the 16S primer sets.”

Additions to the Discussion:

Our response: We very much welcomed these suggestions to extend the discussion, and have implemented them. To maintain clarity in the text structure we have now included Discussion headings.

Examples of Cephalopoda regional diversity

As a novice in understanding localized cephalopod diversity, I wonder if a better approach would be to use many species-specific qPCR assays, rather than an 18S metabarcoding approach. Therefore, providing some information on cephalopod biodiversity hotspots and the number of species in a localized region, would provide the reader with the need for a metabarcoding primerset.

Our response: We appreciate the need for more ecological context about cephalopod biodiversity, and have included this in the discussion. In summary, several studies of local cephalopod diversity report 32 – 85 species, therefore doing many species-specific qPCR assays would be a lot of work and would require detailed pre-existing knowledge of the species composition, which is rarely the case.

Multi-marker approach.

Previous eDNA studies have analyzed metabarcoding data with a multi-marker approach to improve species detection (such as Evans et al. 2017, Li et al. 2018), and I think you need to add additional information about these studies when discussing a mutlimarker cephalopod approach with 18S and 16S.

- Evans, N. T., Li, Y., Renshaw, M. A., Olds, B. P., Deiner, K., Turner, C. R., ... & Pfrender, M. E. (2017). Fish community assessment with eDNA metabarcoding: effects of sampling design and bioinformatic filtering. *Canadian Journal of Fisheries and Aquatic Sciences*, 74(9), 1362-1374.
- Li, Y., Evans, N. T., Renshaw, M. A., Jerde, C. L., Olds, B. P., Shogren, A. J., ... & Pfrender, M. E. (2018). Estimating fish alpha-and beta-diversity along a small stream with environmental DNA metabarcoding. *Metabarcoding and Metagenomics*, 2, e24262.

Our response: We have extended the discussion about the multi-marker approach as suggested.

Discrepancies in detectability between nuclear and mitochondrial eDNA.

Furthermore, with discussion about a multimarker 18S and 16S combination, I think it is imperative that you mention possible discrepancies in detectability between nuclear and mitochondrial eDNA.

- Bylemans, J. et al. Does size matter? An experimental evaluation of the relative abundance and decay rates of aquatic environmental DNA. *Environ. Sci. Technol.* 52, 6408–6416 (2018). <https://doi.org/10.1021/acs.est.8b01071>
- Jo, T. et al. Estimating shedding and decay rates of environmental nuclear DNA with relation to water temperature and biomass. *Environ. DNA*, 2, 140–151 (2020). <https://doi.org/10.1002/edn3.51>

- Moushomi, R. et al. Environmental DNA size sorting and degradation experiment indicates the state of *Daphnia magna* mitochondrial and nuclear eDNA is subcellular. *Sci. Rep.* 9, 1–9. (2019). <https://doi.org/10.1038/s41598-019-48984-7>

Our response: We have now mentioned the possible differences in detectability of nuclear and mitochondrial genes in the paragraph discussing the multi-marker approach.

Figures: I do not see any figure legends.

Our response: We sincerely apologize for this omission. The files were in fact uploaded to the submission system, but did not appear in the rendered PDF. We have made certain all information is now available upon revision, and we look forward to your feedback.

Additionally, we have changed the colours in the figures to also be suitable for colour-blind readers.

Reviewer: 2

Comments to the Author(s)

This is an interesting manuscript that contributes a new primer set to analyse for cephalopods in eDNA samples. While there are still many unknowns involving eDNA processes, this primer set will be helpful as it will allow for cephalopods to be included in the analyses. I have only a few comments/edits here to address:

1. In text citations: should these be in chronological order? Most journals require that, please check this journal to align with their guidelines.

Our response: We have now adjusted all references to the Vancouver reference style as requested by the Editors.

2. Lines 81-86: comment: there needs to be extreme caution with publications stating new ranges, etc using eDNA methods. Just my opinion as I read this section....

Our response: We agree that ecological interpretations from eDNA results should be made with the limitations of the method in mind, for example the limitations of a primer as we describe in this paper. The authors of the cited paper are indeed careful in their interpretation, as reflected by our phrasing "...suggested that the species' distribution might extend further [...] than previously thought",

3. Comment: I think it strengthens your paper by having the empirical testing piece. There are many, many incorrect sequences found in Genbank which could skew your findings. (relating to lines 265-305).

Our response: We agree that empirical testing of a primer set adds to the extent of confidence one can have in the interpretation of results, and hope that this paper highlights the necessity of this practice.

Lines 50-52: Oddly worded sentence, please reword to clarify this statement.

Our response: We hope our rephrasing clarified our argument.

“Additional difficulties in sampling cephalopods result from their possibly patchy distribution and their agility which allows them to avoid or escape sampling gear.”

Line 85: delete “not yet” and replace with “none specifically”

Our response: We have adjusted this sentence as suggested.

Line 307: of the 107 in Genbank- were they all reliably identified? Many readers might be skeptical of the use of Genbank but I absolutely understand the need to use it as it's the best ref we have currently for these types of studies.

Our response: The reliability of these sequences can only really be reviewed by assessing the expertise of the person who conducted the morphological identification, and the method used for sequencing. We did not review the reliability of the individual sequences, and assumed them all correct for this study. However, a comparison between the 107 sequences in GenBank and the sequences in the curated SILVA database shows major overlap, and also the calculated B_c and B_s values are similar. We have added a new paragraph to the discussion where we review possible explanations for discrepancies between our reference sequences and sequences present in GenBank, also in response to a comment by Reviewer 1 (to Lines 280-290).

Conclusions: It is great that additional, reliable sequences are being added to Genbank. I think there is still quite a way to go before eDNA studies are reliable but for using this primer set as part of the tools to study biodiversity and locations of cephalopods will be an aid to the field.